# Extending Density Phase-Field Simulations to Dynamic Regimes

David Jacobson [1] , Reza Darvishi Kamachali [2] and Gregory Bruce Thompson [1,*]

1  Department of Metallurgical and Materials Engineering, University of Alabama,
   Tuscaloosa, AL 35487, USA; dwjacobson@crimson.ua.edu
2  Federal Institute for Materials Research and Testing, 12205 Berlin, Germany; reza.kamachali@bam.de
*  Correspondence: gthompson@eng.ua.edu

**Abstract:** Density-based phase-field (DPF) methods have emerged as a technique for simulating grain boundary thermodynamics and kinetics. Compared to the classical phase-field, DPF gives a more physical description of the grain boundary structure and chemistry, bridging CALPHAD databases and atomistic simulations, with broad applications to grain boundary and segregation engineering. Notwithstanding their notable progress, further advancements are still warranted in DPF methods. Chief among these are the requirements to resolve its performance constraints associated with solving fourth-order partial differential equations (PDEs) and to enable the DPF methods for simulating moving grain boundaries. Presented in this work is a means by which the aforementioned problems are addressed by expressing the density field of a DPF simulation in terms of a traditional order parameter field. A generic DPF free energy functional is derived and used to carry out a series of equilibrium and dynamic simulations of grain boundaries in order to generate trends such as grain boundary width vs. gradient energy coefficient, grain boundary velocity vs. applied driving force, and spherical grain radius vs. time. These trends are compared with analytical solutions and the behavior of physical grain boundaries in order to ascertain the validity of the coupled DPF model. All tested quantities were found to agree with established theories of grain boundary behavior. In addition, the resulting simulations allow for DPF simulations to be carried out by existing phase-field solvers.

**Keywords:** density phase field; grain boundaries; numerical methods





## 1. Introduction

Grain boundaries play an out-sized role in determining material properties in metals such as strength, electrical and thermal conduction, etc. A significant amount of research has been conducted to determine how particular grain boundary arrangements can be achieved through different synthesis processes and conditions in order to obtain desired properties in polycrystalline materials. A more difficult problem has been determining how to predict and control the evolution of grain boundary networks over time. The ability to accurately model the evolution of grain boundaries is necessary for being able to predict how metallic material properties evolve in response to heat, stress, magnetic fields, and other phenomena [1]. Such modelling has been facilitated by the advent of high-performance computer systems which now alloww for in silico experiments to extend from the quantum scale to system/macro level simulations.

Many of today's open problems in materials science have to do with limitations to computational modelling associated with length and timescales. The larger the system modelled, the more underlying physics is often neglected or simplified in order to make the problem computationally feasible. The problem of grain boundary network evolution falls into such a category. Classical phase-field methods have emerged on the mesoscale describing microstructure evolution [2]. Significant progress has been made in phase-field modeling of grain boundary motion in polycrystalline materials, especially with the development of the multi-phase-field methods to explore grain boundary junctions,

vertexes and their dynamics in various setups [3–5]. Yet, grain boundaries exhibit an incredible level of detail and variation in behavior at the atomic scale (low-angle grain boundary arrangements, special character boundaries, random high-angle boundaries, twist vs. tilt configurations, etc.). Atomistic simulations can capture such details of many if not all of these grain boundary behaviors but are computationally too expensive for studying the influence of such phenomena on the time scales of grain boundary network coarsening. The classical phase-field simulations sacrifice much of the aforementioned details of grain boundary physics in order to operate at lengths and timescales at which grain boundary coarsening can be observed, relying on phenomenological descriptions of grain boundaries as opposed to physical models. An overview of the application of phase-field methods to grain growth problems is given in the following references [6–10]. A broader overview of traditional phase-field theory and capabilities can be found in the following references [3,4,11,12].

To overcome some of these challenges, density phase-field (DPF) methods have arisen as an attractive alternative for describing grain boundaries in a more physical manner than traditional phase-field methods based on logistic-type order parameters ($\phi$) [13–18]. The idea behind DPF simulations is that the free energy of a grain boundary is more directly related to the atomic density at the grain boundary as opposed to traditional order parameters. The reduced density and disordered atomic environments at grain boundaries result in significant bond strains that are the ultimate source of the excess grain boundary energy. The atomic density order parameter enables DPF simulations to be integrated with CALPHAD databases while strongly linked with atomistic simulations in a physically-sound manner. The resulting density-based free energy functional is able to capture the temperature and composition dependence of grain boundary energetics, giving an accurate description of grain boundary phase behavior that may not present in traditional phase-field simulations [19,20]. The remainder of this introduction provides a brief overview of the types of DPF simulations available, as well as a discussion on the current technical challenges associated with carrying out DPF simulations.

### 1.1. Overview of Density Phase-Field Theory

DPF solvers work similarly to their classical counterparts in that they minimize a free energy functional of the form given by Equation (1) by solving a model A type equation for unconserved dynamics (Equation (2)).

$$F = \int_v F_v(\rho, \nabla\rho, \dots)\, dV \tag{1}$$

$$\frac{\partial \rho}{\partial t} = -m_\rho \mu_\rho \tag{2}$$

with the potential function $\mu_\rho = \frac{\delta F_v}{\delta \rho}$. Typically, a normalized form of atomic density is used such that $\rho = 1$ at equilibrium associated with the parent bulk phase. One can calculate the real atomic density simply by multiplying the reference bulk atomic density being studied ($\rho^0_{atom}$) by the local value of the normalized density ($\rho_{atomic} = \rho^0_{atomic}\rho$). To avoid confusion between $\rho$ and $\rho_{atom}$, the reference atomic density is expressed as the inverse of the molar volume ($\rho_{atom} = \hat{V}_0^{-1}$). The form of the volumetric free energy function $F_v$ varies between models. In its most basic and general form, it mirrors classical phase field free energy functionals in that it is composed of two terms: a bulk term to describe the free energy of the system as a function of local density plus a gradient energy series to take into account the energy associated with spatial variations of $\rho$. The general form of the volumetric free energy functional is given below.

$$F_v = f_{bulk}(\rho) + \sum_i \kappa_i |\nabla^i \rho|^2 \tag{3}$$

Kamachali derived the volumetric free energies as a deviation from the bulk free energy curve of a solid solution [13]. For a regular solution, the free energy is presented in Equation (4)

$$F_v = X_A \left( E_A^B \rho^2 + (K_A^B + PV_A - TS_A^B)\rho + \kappa_{A,1}|\nabla\rho|^2 + \kappa_{A,2}|\nabla^2\rho|^2 \right)$$
$$+ X_B \left( E_B^B \rho^2 + (K_B^B + PV_B - TS_B^B)\rho + \kappa_{B,1}|\nabla\rho|^2 + \kappa_{B,2}|\nabla^2\rho|^2 \right)$$
$$+ \rho^2 \Omega X_A X_B - T\Delta S_{mix}^B + \kappa_X |\nabla X_B|^2 \quad (4)$$

Equation (4) resembles the standard CALPHAD formulation, upgraded with the density-dependent terms and the gradient energy terms. Compared to the classical phase-field models [3], a major advantage of DPF approach is that it allows for the natural development of grain boundary free energy functional, integrated with the CALPHAD framework. This allowed for the successful prediction of spinodal decomposition occurring at the grain boundaries outside of the bulk miscibility gap in the iron alloys [14,19] as well as in the platinum-gold system [15]. Jacobson et al. took a different approach by using atomistic theory to derive a free energy functional based on interatomic potentials [21]. The resulting class of simulations has been termed the Molecular Phase-Field method (MoPF); the general and Morse forms of which are given by Equations (5) and (6).

$$F_v = \frac{\rho}{\hat{V}_0} \sum_i \frac{n_i}{2} U_{bond}(\rho) + \sum_i \kappa_i |\nabla^i \rho|^2 \quad (5)$$

$$F_v = \frac{\rho}{\hat{V}_0} \sum_i \frac{n_i}{2} \epsilon \left[ e^{-2\alpha(r_i^* \rho^{-\frac{1}{3}} - r_0)} - 2e^{-\alpha(r_i^* \rho^{-\frac{1}{3}} - r_0)} \right] + \sum_i \kappa_i |\nabla^i \rho|^2 \quad (6)$$

The primary advantage of the MoPF model is that the interatomic potential parameters used as inputs to the model naturally convey material-specific characteristics to the model. For example, in the absence of gradients, the MoPF method correctly predicts the bulk modulus of the material. For grain boundaries, the grain boundary free energy is a natural consequence of the model rather than an objective value that the model must be calibrated for in order to reproduce. However, the atomistic simulations can be computationally expensive.

There are two key differences between classical phase-field and DPF methods. In a classical phase-field simulation, the order parameter $\phi$ represents crystallographic misorientation, but the association between the misorientation and the order parameter is rather arbitrary. The atomic density field on the other hand is physically linked with the substructure of the grain boundary. This constraint is taken into account by modifying the model A equation. Typically, the model A equation relies on a potential function $\mu_\rho$ that follows a non-conserved variational form given below.

$$\mu_\rho = \frac{\delta F}{\delta \rho} = \frac{\partial F_v}{\partial \rho} - \nabla \frac{\partial F_v}{\partial \nabla \rho} \quad (7)$$

Considering the mass conservation, the evolution of the density field shall be taken as below, taking $\Delta\mu_\rho$ instead of $\mu_\rho$.

$$\frac{\partial \rho}{\partial t} = -m_\rho \Delta\mu_\rho \quad (8)$$

Here, the potential difference indicates the relation between the change in density and mass transfer into or out of the boundary. Assumed in the density-based model is that the source/sink of these atoms corresponds to a reservoir consisting of a perfect crystal at constant density ($\rho = 1$) where the density potential $\mu_\rho = \mu_\rho^0$. In such a reservoir, the gradient contribution to the density potential is zero. If we assume that the energy change due to a change in reservoir volume is negligible (a change in reservoir volume is necessary

to keep the density constant), the resulting form of $\mu_\rho^0$ is simply the volumetric free energy of an unstrained perfect crystal as is shown in Equation (9).

$$\Delta\mu_\rho = \frac{\delta F}{\delta \rho}(\rho, \nabla\rho, ...) - F_v(\rho = 1, \nabla\rho = 0, ...) \tag{9}$$

In reality, density changes in a grain boundary typically occur through vacancy emission and absorption [22]. The reservoir concept defined above is meant to provide a simpler and more computationally efficient means of modelling grain boundary density dynamics than attempting to model vacancy migration, generation, and elimination at the mesoscale.

Another key difference between the DPF methods and classical phase-field simulations deals with what we term the center boundary condition. Grain boundaries are non-equilibrium defects that are inherently unstable. If not for the existence of a large activation barrier associated with grain rotation, grains would simply reorient themselves such that all grain boundaries were eliminated. The aforementioned activation barrier is a direct result of the crystallographic misorientation between the two grains composing the boundary. Unfortunately, this activation barrier is not captured in DPF models which necessitates a "center boundary condition", i.e., the density at the grain boundary center must be specified and held constant throughout the simulation. If this was not the case, grain boundaries in DPF simulations would simply dissipate until the normalized density everywhere equaled one (the equilibrium value). The density at the center boundary condition is the lowest density in the entire simulation and is referred to as $\rho_{min}$. The choice of $\rho_{min}$ can be based on a variety of criteria but is most often meant to relate with the misorientation angle [13] and to match the minimum calculated density of a grain boundary generated using molecular dynamics.

### 1.2. Issues with the Density Phase-Field Method

#### 1.2.1. Theoretical Issues

The DPF model was derived assuming that only attractive interatomic forces were needed. By doing so, this greatly simplifies the complexity of calculating $\mu_\rho$, but at the expense of thermodynamic consistency. We illustrate this problem using the single component free energy functional and its respective density potential difference given below.

$$F_v = \rho(\rho\hat{E}_A + \hat{K}_A + P\hat{V}_A - T\hat{S}_A) + \kappa_\rho|\nabla\rho|^2 \tag{10}$$

$$\Delta\mu_\rho = (2\rho - 1)\hat{E}_A - \kappa_\rho\nabla^2\rho \tag{11}$$

In the absence of any gradients, $\Delta\mu_\rho(\rho = 1)$ should equal zero because $\rho = 1$ is defined as the equilibrium condition. Due to the simple linear form of the potential energy, this condition is not naturally met and is imposed by the condition of having $\rho \leq 1$ throughout the system. While being a robust solution for studying static grain boundaries, we have found that such enforcement schemes introduce non-physical behaviors into dynamic simulations. This motivated our desire to improve the density-based free energy functional such that the equilibrium condition is met in a dynamic state.

#### 1.2.2. Computational Challenges

There is a computational challenge associated with the center boundary condition. Although the center boundary condition is necessary for grain boundaries to exist in DPF simulations, it makes simulating the motion of grain boundaries difficult. The question arises as to how can the center boundary condition be moved such that the density field evolves in a natural manner? Although distinct from DPF, the authors point to the work by Phillippe et al. to further illustrate the numerical difficulties that arise from solving the evolution equations of grain boundaries using non-trivial free energy functionals [23].

The gradient energy terms in the DPF free energy functional are another source of difficulty in carrying out DPF simulations. Classical phase-field models are limited to a single first-order term in gradient energy. However, higher-order gradient terms are known to sig-

nificantly improve the accuracy of phase-field models as well as the DPF model. Additionally, the resulting higher-order partial differential equations (PDEs) are much stiffer and more computationally intensive to solve than lower-order PDEs. The DPF simulations are shown to give a smooth density profile with a minimum of first- and second-order gradient terms. This fact is demonstrated in Figure 1, one computed with only the first-order gradient term (plot A) and another with both first- and second-order terms (plot B). As a result, the model A equation for DPF simulations is a fourth-order PDE that is significantly more difficult and time-consuming to solve numerically than it is for classical phase-field simulations.

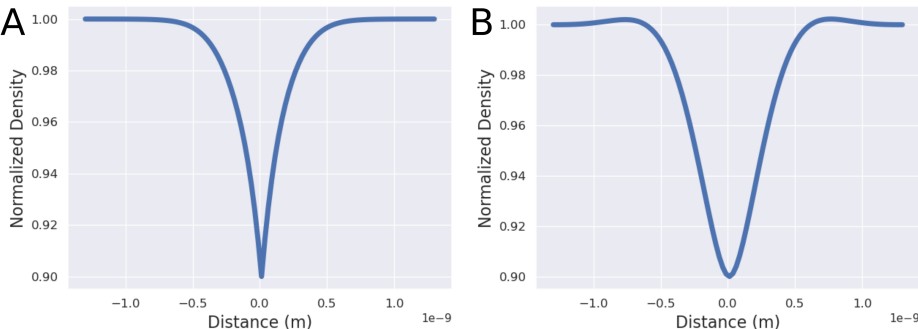

**Figure 1.** Plot (**A**): the density field of a non-coupled DPF simulation where only the first term of the gradient energy series is included. Plot (**B**): the density field of a non-coupled DPF simulation where the first two terms of the gradient energy series are included. Notice that the inclusion of the higher-order gradient energy term eliminates the sharp point at the grain boundary center. The inclusion of the of the higher-order gradient energy terms also results in the larger than bulk density regions at the grain boundary periphery.

In this study, we seek to address the issues outlined above and answer the following questions:

- What are the thermodynamic criteria that density phase field free energy functionals should meet?
- How can the DPF methods be made dynamic?
- How can the performance constraints associated with solving a fourth-order PDE be overcome?

## 2. Theory

### 2.1. General Criteria for Density Free Energy Functionals

This section is dedicated to answering question one from the introduction. Generally, a thermodynamically consistent density free energy functional needs to, at a minimum, meet the following criteria:

1. $F_v(\rho = 1, \nabla\rho = 0) = F_v^0$
2. $\Delta\mu_\rho(\rho = 1, \nabla\rho = 0) = 0$
3. $\frac{\partial\mu_\rho}{\partial\rho}(\rho = 1, \nabla\rho = 0) > 0$
4. $F_v(\rho = 0, \nabla\rho = 0) = 0$
5. $\frac{\partial F_v}{\partial\rho}(\rho = 0, \nabla\rho = 0) = 0$

The first of these criteria is the continuity condition stating that the free energy should align with the bulk value $F_v^0$, whenever the excess density and its gradients vanish. The second criteria ensures that the bulk state of ($\rho = 1, \nabla\rho = 0$) is a true equilibrium state. The third criteria ensures that the bulk equilibrium is stable. Criteria 4 and 5 are referred to as the sparse particle conditions. They ensure that atoms that are far apart have a negligible influence on one another. It should be noted that these criteria are useful for making the volumetric free energy more physically accurate across its entire domain, but have limited effect on the region of the free energy curve ($\rho \approx 1$) relevant to grain boundary simulations.

The original density-based formulation does not meet the second and third criteria because repulsive interatomic contributions were not included. As noted above, such interactions can be explicitly modelled using the approach taken by Jacobson et al. by constructing the volumetric free energy functional from interatomic potentials. Here, we take a simpler approach by deriving a polynomial from of the bulk component of the volumetric free energy that satisfies criteria 1–5. For a grain boundary at $T = 0$ K, the kinetic and entropic portions of the free energy functional are zero. We further assume that the pressure volume contribution is small enough to be neglected. Thus, the volumetric free energy can be expressed purely in terms of the molar potential energy $\hat{E}_A$ ($F_{v,bulk} = \frac{\rho}{\hat{V}_0}\hat{E}_A(\rho)$.) Recognizing that $\hat{E}_A(\rho = 1)$ is the cohesive energy ($E_{co}$), one can write

$$F_{v,bulk} = \rho E_{co} f(\rho) \tag{12}$$

Equation (12) meets criteria 1–3 when $f(\rho)$ gives $f(\rho = 1) = 1$, $\frac{\partial f}{\partial \rho}|_{\rho=1} = 0$ and $\frac{\partial^2 f}{\partial \rho^2}|_{\rho=1} < 0$ as $E_{co} < 0$. Infinitely many different polynomial forms can be constructed to meet such criteria. Under the premise that simpler is better, we use the following form of $f(\rho)$.

$$f(\rho) = \rho^n(a\rho^2 + b\rho + c) \tag{13}$$

Parameters $n$ and $a$ are free variables that can be used to change the potential well shape which corresponds to a change in the bulk properties of the material being modelled. The characteristic shape of Equation (13) is shown in Figure 2 where the curves illustrated demonstrate the influence of parameters $a$ and $n$ on the well shape. Once parameters $n$ and $a$ are chosen, $b$ and $c$ can be determined through the following equations.

$$b = -(n + 2a) \tag{14}$$

$$c = 1 + a + n \tag{15}$$

A more thorough derivation of the above two expressions is provided in the Appendix A.

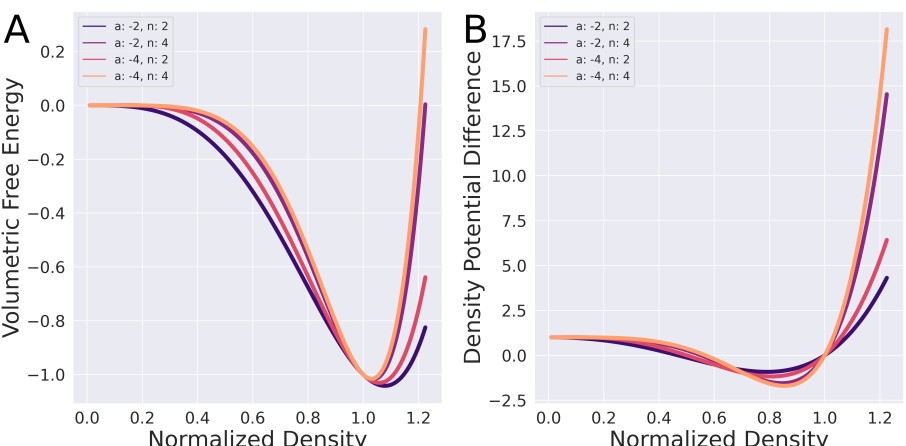

**Figure 2.** Plot (**A**): the bulk volumetric free energy function from Equations (12) and (13). Plot (**B**): the bulk component of the density potential difference corresponding to to Equations (12) and (13). Values of the cohesive energy and bulk modulus are $-1$ and $1$, respectively ($E_{co} = -1$, $\hat{V}_0^{-1} = 1$). Parameters $a$ and $n$ are varied to show their influence on the free energy curve well shape. Notice that for every curve the density potential difference at a density of one equals zero. This is by design to ensure that the bulk state ($\rho = 1$) is a stable equilibrium.

### 2.2. Linking Order Parameters with Density

Questions two and three can be solved by coupling the density phase-field method with traditional order parameters. We reintroduce the order parameter $\phi$ into the simulation

as an independent field variable and make the density dependent on $\phi$ according to Equation (16)

$$\rho = 1 - 4(1 - \rho_{min})\phi(1 - \phi) \tag{16}$$

Doing so resolves the issues of the center boundary condition because the density field is naturally at a minimum where $\phi = 0.5$. Now the temporal evolution of the density variable can be achieved by solving the model A equation with respect to $\phi$ and recalculating $\rho$ using Equation (16) after every time step. The evolution equation is derived below.

$$\frac{\partial \phi}{\partial t} = m_\phi \left( \frac{\delta F}{\delta \phi} - \mu_\phi^0 \right) \tag{17}$$

$$\frac{\delta F}{\delta \phi} = \frac{\partial F_v}{\partial \phi} - \nabla \frac{\partial F_v}{\partial \nabla \phi} \tag{18}$$

The reference component of Equation (17) ($\mu_\phi^0$) can be calculated simply by appending the derivative of $\rho$ with respect to $\phi$ to the equivalent reference component in Equation (9).

$$\mu_\phi^0 = F_v(\rho = 1, \nabla \rho = 0)\frac{\partial \rho}{\partial \phi} \tag{19}$$

The first term of Equation (18) can be solved for through use of the chain rule.

$$\frac{\partial F_v}{\partial \phi} = \frac{\partial F_v}{\partial \rho} \frac{\partial \rho}{\partial \phi} \tag{20}$$

A direct connection between the mobilities ($m_\phi$ and $m_\rho$) is not possible because the use of the center boundary condition when carrying out density dynamics prevents the expression $\frac{\partial \rho}{\partial t} = \frac{\partial \rho}{\partial \phi} \frac{\partial \phi}{\partial t}$ from being valid over the entire domain. The second term of Equation (18) can be solved for explicitly by substituting Equation (16) into the gradient free energy series. Doing so increases the complexity of the free energy functional though without alleviating the large performance constraints associated with solving a fourth-order PDE. Instead, we motivate an approximation that simplifies the free energy functional in addition to making the evolution equation second order.

　　The gradient energy term is meant to take into account the excess energy resulting from non-equilibrium environments associated with the spatial transition of a field variable. For the case of DPF simulations, the gradient energy is meant to take into account the disordered bonding environment found at grain boundaries. Atomistic simulations indicate that the greatest degree of disorder is found at the center of the grain boundary as opposed to at its periphery. Thus, we can surmise that the gradient energy series should predict a maximum free energy at the grain boundary center. This is why gradient energy terms with orders in excess of one are required for non-coupled DPF simulations. At the grain boundary center, the density gradient must be zero and by extension the first order term of the gradient energy series is zero. Even when a second order term is included, there are still non-physical artifacts in the resulting density field. One will notice in Figure 1 that the sharp grain boundary center is eliminated by the inclusion of the second order term, but that the density is now overestimated in the periphery of the grain boundary where the density values exceed 1. It is possible that the addition of more higher-order terms would eventually yield a satisfactory density profile, but determining the coefficients for said terms would be non-trivial and the resulting PDEs would be stiff and require time step sizes that are too small to be considered practical. As a result, we make the following simplifying assumption.

$$\sum_i \kappa_i |\nabla^i \rho|^2 \approx \kappa_\phi |\nabla \phi|^2 \tag{21}$$

The motivation for this assumption is two-fold. From a computational stand point, it reduces the evolution equation to second order and simplifies the free energy functional.

From a theoretical stand point, we simply note that the density profiles obtained using the approximation given by Equation (21) more closely match density profiles obtained from atomistic simulations than density profiles obtained using the the density gradient energy sum. Examples of density profiles obtained from atomistic simulations are shown in the second figure of the work by Jacobson et al. [21]. The fully coupled form of the volumetric free energy functional that will be used throughout the remainder of this work is given by Equation (22)

$$F_v = \frac{\rho^{n+1}}{\hat{V}_0} E_{co}(a\rho^2 + b\rho + c) + \kappa_\phi |\nabla \phi|^2. \tag{22}$$

The order parameter and density profiles resulting from a free energy expression, such as Equation (22), are given in Figure 3.

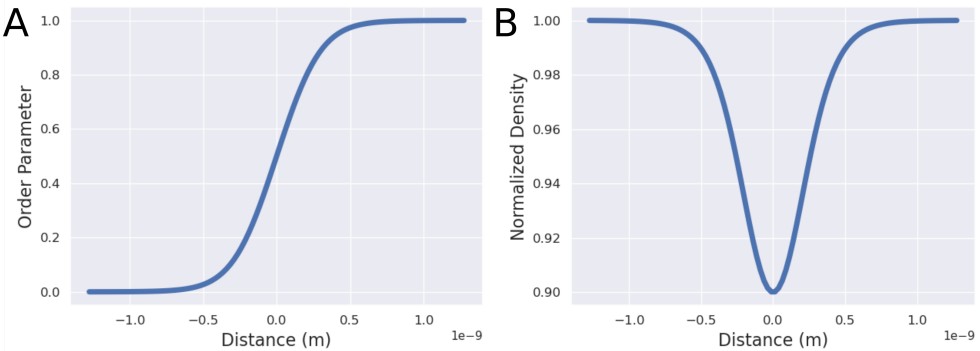

**Figure 3.** Plot (**A**): the equilibrium order parameter field $\phi$ achieved by minimizing a free energy functional with a volumetric free energy of the form given by Equation (22). Plot (**B**): the equilibrium density profile obtained by taking the curve of $\phi$ present in plot (**A**) and using Equation (16) to determine $\rho$.

## 3. Methods

We seek to demonstrate the validity of coupling $\rho$ and $\phi$ in addition to the enhanced functionality afforded by doing so. To this end both one and two dimensional DPF models were developed that use explicit finite differencing to solve the model A equation for the time evolution of the density and order parameter fields. This section outlines a series of tests that are used to validate the use of coupled DPF simulations as well as demonstrate their ability to accurately describe grain boundary physics.

### 3.1. Equilibrium Grain Boundary Properties

The two main equilibrium quantities of interest in grain boundary simulations are the boundary width and excess energy. Real grain boundaries are approximately 1 nm wide with excess free energies on the order of $1 \frac{J}{m^2}$ at room temperature. These are average values under normal conditions with variations possible because of material composition, grain boundary type, temperature, stress state, etc. We evaluate the fitness of the coupled DPF model by varying the values of $\rho_{min}$ and $\kappa_\phi$ in order to see their influence on both grain boundary width and excess energy. The grain boundary energy is calculated using Equation (23).

$$\gamma = \frac{1}{A} \int [F_v - \rho F_v(\rho = 1)] dV \tag{23}$$

The grain boundary width can be calculated from the profile of $\rho$ using a simple threshold criterion, e.g., the grain boundary region consists of the region of the density curve where $\rho < \rho_{cut}$). The expression used for $\rho_{cut}$ is provided below.

$$\rho_{cut} = 1 - 0.01 \left( \frac{1 - \rho_{min}^0}{1 - \rho_{min}} \right) \tag{24}$$

### 3.2. Dynamic Properties of Planar Grain Boundaries

For planar grain boundaries, the intrinsic driving force for grain growth is zero. A number of non-intrinsic driving forces for grain growth can be imposed on planar grain boundaries such that grain boundary motion occurs. A linear relationship exists between the boundary velocity and the driving force per unit area (pressure) as shown in Equation (25) where $v$ equals the grain boundary velocity, $m_{GB}$ equals the grain boundary mobility, and $P$ equals the driving pressure for motion.

$$V_{GB} = m_{GB}P \tag{25}$$

In order for a phase-field model to be considered valid, Equation (25) should hold. More explicitly, the relationship between grain boundary velocity and driving force should be linear. The means by which an applied driving force can be exerted on a planar grain boundary in a phase-field simulation is shown below.

The relation between grain boundary velocity and the evolution of the field variable $\phi$ is given by Equation (26).

$$v_{GB} = \dot{\phi}|\nabla\phi|^{-1} \tag{26}$$

Combining Equations (25) and (26), one can express the evolution of the order parameter field in terms of the applied driving force.

$$\dot{\phi} = m_{GB}P|\nabla\phi| \tag{27}$$

The grain boundary mobility multiplied by the applied driving pressure we refer to as the speed factor. Finally, Equation (27) can be added to the model A equation (Equation (17) to provide a motion equation for the order parameter field that includes the influence of an external driving force for grain growth.

$$\dot{\phi} = -m_\phi\mu_\phi + m_{GB}|\nabla\phi|P \tag{28}$$

### 3.3. The Shrinking Circular Grain Problem

The shrinking circular grain problem is a useful means of evaluating the accuracy of phase-field methods because it can be compared with an analytical solution. We describe the mathematics surrounding the problem here as well as the method by which the order parameter mobility can be determined using the shrinking circular grain problem. The velocity of a grain boundary can be expressed as a mobility times a pressure.

$$V_{GB} = m_{GB}P \tag{29}$$

If we consider the case of a 2D circular grain with radius $r$, the intrinsic driving pressure can be expressed in terms of the grain boundary energy and the grain radius.

$$P_{intrinsic} = \frac{2\gamma}{r} \tag{30}$$

For a grain boundary that is moving with constant velocity parallel to the grain boundary surface normal, the forward motion of the grain boundary can be related to the local rate of change in the order parameter $\phi$ using the equations below.

$$V_{GB} = \frac{\dot{\phi}}{|\nabla\phi|} \tag{31}$$

$$\frac{2m_{GB}\gamma}{r} = -\frac{1}{|\nabla\phi|}\left(m_\phi\Delta\mu_\phi\right) \tag{32}$$

The mobility of the $\phi$ field can be obtained by rearranging this equation into the following form.

$$m_\phi = -\frac{2m_{GB}\gamma|\nabla\phi|}{r\Delta\mu_\phi} \tag{33}$$

The grain boundary mobility and excess free energy can be determined using molecular dynamics [24–28]. At the grain boundary center ($\phi = 0.5$), the bulk and reference terms of the density potential difference are zero such that $\Delta\mu_\phi = -2\kappa_\phi\nabla^2\phi$. In order to solve for $\nabla\phi$ and $\nabla^2\phi$, we assume the order parameter field can be accurately approximated using a logistic function.

$$\phi = \frac{1}{1 + e^{-k(x-x_0)}} \tag{34}$$

In polar coordinates, the gradient and Laplacian of $\phi$ are

$$\nabla\phi = \hat{e}_r k\phi(1 - \phi) \tag{35}$$

$$\nabla^2\phi = \frac{\partial^2\phi}{\partial r^2} + \frac{1}{r}\frac{\partial\phi}{\partial r} \tag{36}$$

If we evaluate the Laplacian of $\phi$ at $\phi = 0.5$ we obtain the following expression.

$$\nabla^2\phi = \frac{|\nabla|}{r} \tag{37}$$

Substituting back into Equation (33) we obtain the following relationship for the mobility of $\phi$.

$$m_\phi = \frac{2m_{GB}\gamma}{\kappa_\phi} \tag{38}$$

### 3.4. Free Energy Functional Parameterization

A number of parameters must be determined in order to simulate grain boundary motion through the minimization of Equation (1) with Equation (22) used as the volumetric free energy. All relevant parameters are listed in Table 1.

**Table 1.** Density phase-field simulation parameters.

| Symbol | Value | Description | Units |
|---|---|---|---|
| $E_{co}$ | $4.32 \times 10^5$ | The cohesive energy | $\frac{J}{mole}$ |
| $\hat{V}_0$ | $6.6 \times 10^{-6}$ | The equilibrium molar volume | $\frac{m^3}{mole}$ |
| $n$ | 4 | Free energy parameter | none |
| $a$ | $-4$ | Free energy parameter | none |
| $\kappa_\phi$ | $1.5 \times 10^{-14}$ | Gradient Energy coefficient | $\frac{J}{m}$ |
| $m_\phi$ | $6.5 \times 10^6$ | order parameter mobility | $\frac{1}{Pas}$ |
| $\rho_{min}$ | 0.9 | minimum grain boundary density | none |

The cohesive energy energy and equilibrium molar volume correspond to nickel. Free energy parameters were chosen because they satisfy satisfy thermodynamic criteria 1–5 and give rise to ideally shaped free energy curves. A "typical" value of $\rho_{min}$ was selected for grain boundaries studied in the Olmsted database and does not correspond to a specific grain boundary [29,30]. The value of $\kappa_\phi$ was set so that in combination with the other thermodynamic criteria the grain boundary width would equal one. The order parameter mobility was calculated using Equation (38). We assume that the grain boundary mobility is $100\frac{m}{sGPa}$. In the same vein as for the choice of $\rho_{min}$, a mobility of $100\frac{m}{sGPa}$ is a typical values for nickel grain boundaries as calculated in the study by Olmsted [29,30]. The authors emphasize here that the good agreement shown between our work (see results section)

and the work performed by Olmsted is indicative that our modifications to the DPF model are valid.

## 4. Results

### 4.1. Equilibrium Results

The equilibrium one dimensional order parameter and density profiles associated with the coupled DPF free energy functional are presented in Figure 3. As mentioned in the theory section, the density profiles associated with the coupled form of the volumetric free energy function results in more accurate density profiles in comparison with atomistics than does the non-coupled volumetric free energy functional. In particular, the discontinuity in the first derivative of the density curve is eliminated at the grain boundary center without the "shoulder regions" and higher computational overhead that results from the inclusion of higher-order gradient energy terms.

If the gradient energy series is truncated to a single term, the relationship between grain boundary width and the gradient energy coefficient should follow a square root relationship [13]. Figure 4 demonstrates that such a relationship is obeyed for physically relevant grain boundary widths.

### 4.2. Dynamic Results

Using the coupled free energy expression, the density description of grain boundaries can be made mobile. For the steady state case of a grain boundary moving at constant velocity, Equation (25) has been shown to hold both experimentally and using molecular dynamic simulations [29,30]. Figure 5 illustrates that coupled density phase-field simulations also obey this trend.

For transient problems, we use the classic 2D phase-field problem of a shrinking circular grain to study the motion of coupled DPF boundaries. The analytical relation between the grain radius and time is given by Equation (39) [5].

$$r = \sqrt{r_0^2 - 4m_\phi \kappa_\phi t} \tag{39}$$

In Figure 6 one can see the comparison of the analytical solution with the numerical solution of a shrinking 4 nm grain. The numerical solution is nearly identical to the analytical solution, expressing both the correct magnitude and trend of the radius vs. time curve. Assumed in Equation (39) is that the radius of the spherical grain is appreciably larger than the width of the interface, thus the slight deviation of the numerical solution from the analytical solution for small r values is not surprising.

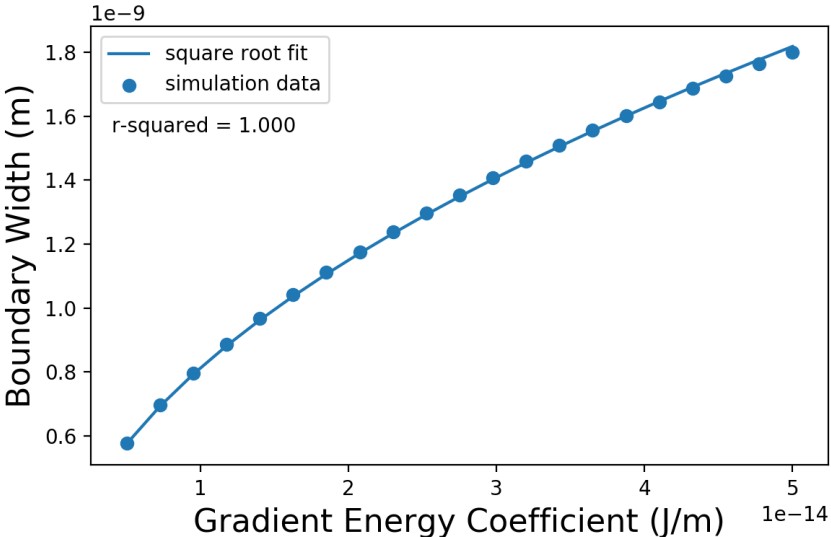

**Figure 4.** Plotted above is the equilibrium grain boundary width vs. the value of the gradient energy coefficient (data points) as well as a line of best fit corresponding to a square root function (the expected theoretical relationship). Visually the agreement between simulation and theory is exemplary and the $R^2$ value of the curve fit is 1 when calculated to three decimal places.

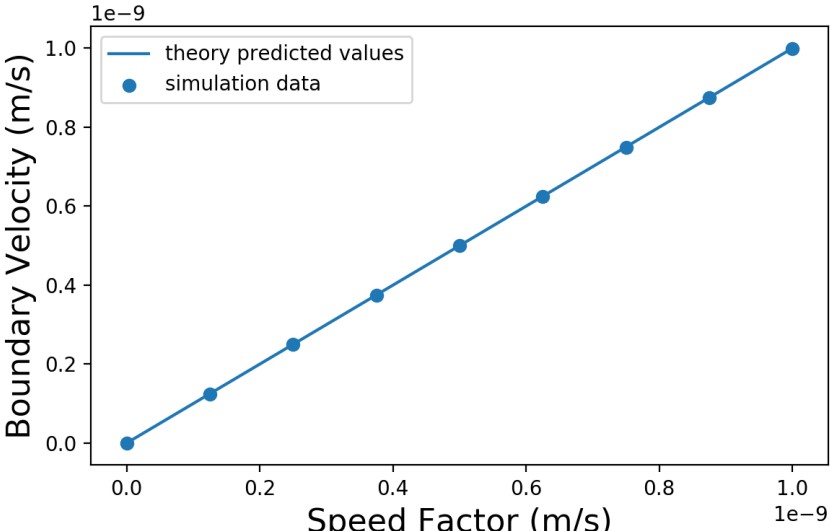

**Figure 5.** Plotted above are steady state grain boundary velocities achieved through the application of a synthetic driving force vs. speed factor (the theoretically predicted grain boundary velocity). It can be seen that there is good agreement between simulation and theory as indicated by the overlapping of the simulation data points and the solid line that represents the linear relationship between velocity and the product of grain boundary mobility and applied driving pressure.

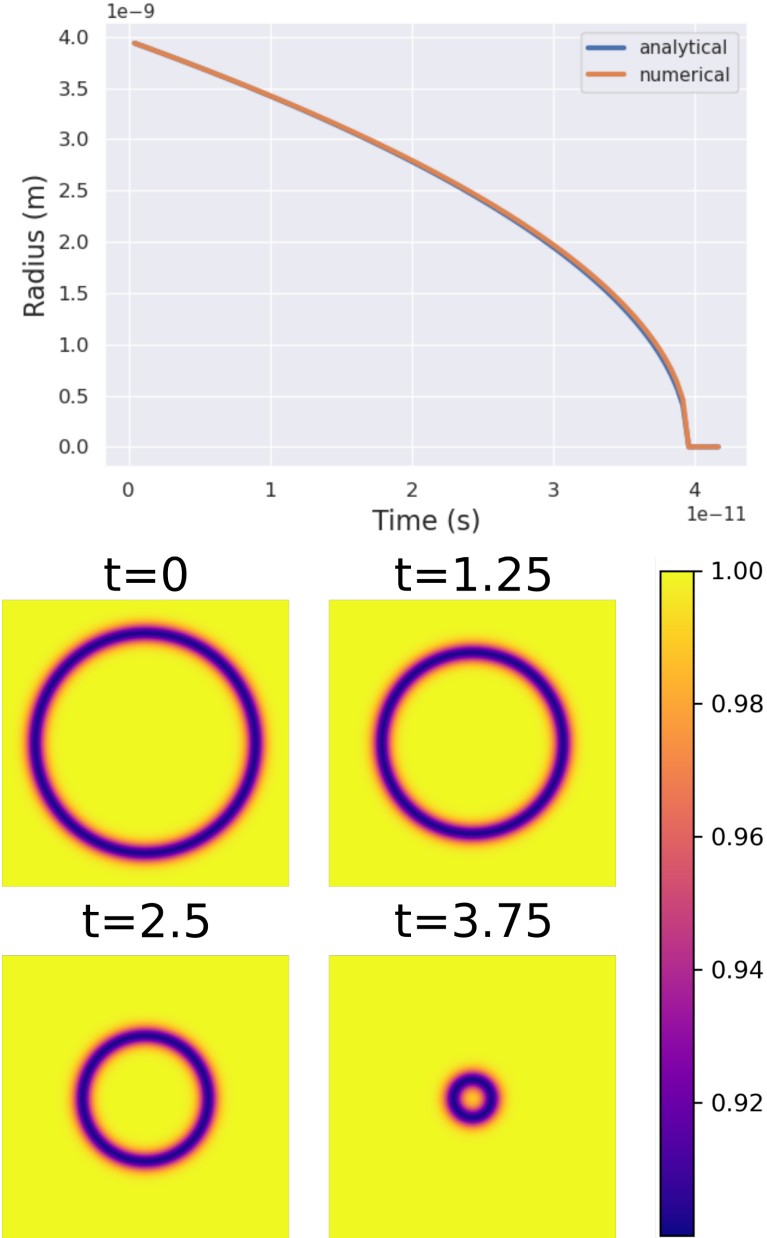

**Figure 6.** The shrinking circular grain problem modelled using the coupled DPF method. Note that the radius vs. time plot exhibits the characteristic parabolic shape. The color scale corresponds to the normalized density field.

## 5. Discussion

With respect to the derivation presented in Section 2.1, the resulting free energy functional has more utility than just meeting the thermodynamic consistency criteria. The well shape of the free energy curves presented in Figure 2 reflects the underlying atomic interactions that give rise to similarly shaped interatomic potential functions, where such a well shape motivated Jacobson to base a DPF free energy functional directly off of interatomic potentials, albeit being theoretically satisfying but computationally expensive. This computational expense is now avoided by using the traditional phase field order parameter constructed with the grain boundary center-line constraints. Equation (13) can easily be manipulated through the *a* and *n* parameters to achieve a well shape very nearly identical to those generated through interatomic potentials without the associated computational overhead. Furthermore, Equation (13) can be extended through the inclusion of more polynomial terms such that additional modification to the well shape can be

achieved. As a result, the correct parameterization of the free energy functional gives rise to an accurate description of grain boundary physics in addition to some macroscale material properties such as bulk modulus.

The primary benefits of coupling the density field with an order parameter is that it provides a means by which to carry out dynamic field simulations in a more computationally efficient manner. An additional advantage that coupled DPF simulations have over "regular" DPF simulations is that expressing $\rho$ in terms of $\phi$ makes it much easier to incorporate DPF methods into traditional phase-field solvers. This back compatibility with existing software will not only make implementation easier, but will also allow for methods that have been used to accelerate classical phase-field simulations to be used to accelerate DPF simulations.

## 6. Conclusions

A set of thermodynamic criteria have been developed that can be used to ensure thermodynamic equilibrium and stability of the bulk state in density-based free energy functionals. The density field variable has been expressed in terms of traditional order parameter type variables in order to make DPF grain boundaries mobile. Approximating the density gradient energy sum in terms of $\phi$ results in a multiple order of magnitude increase in computational performance as well as a more accurate density profile. Finally, the dynamic DPF simulations are shown to be physical by simulating a circular grain whose change in radius with time matches the parabolic analytical solution. The combination of a more accurate density profile and the excellent agreement between the numerical and analytical solutions of the circular grain shrinkage problem indicate that coupling density fields with traditional order parameters in DPF simulations is a sound means by which to make DPF simulations dynamic while also improving simulation performance and accuracy.

**Author Contributions:** D.J. provided conceptualization, methodology, programing, validation, formal analysis, investigation, data curation, and writing the original draft. R.D.K. provided conceptualization, methodology, and review and editing. G.B.T. provided conceptualization, review and editing, supervision, project administration, and funding acquisition. All authors have read and agreed to the published version of the manuscript.

**Funding:** The authors thankfully acknowledge the National Science Foundation (DMR 1709803) for supporting this work. RDK acknowledges the financial support from DFG, project DA 1655/2-1 in the Heisenberg program.

**Data Availability Statement:** Data will be made available upon request to the authors.

**Conflicts of Interest:** The authors declare that they have no known competing financial interest or personal relationships that could have appeared to influence the work reported in this paper.

## Appendix A. Density Polynomial Derivation

### General

$$F_v = \frac{\rho}{\hat{V}} \hat{F} \tag{A1}$$

$$\hat{F}(\rho = 1) = \hat{E}_{co} \tag{A2}$$

### Criteria

1. $\hat{F}(\rho = 1) = E_{co}$
2. $\hat{F}(\rho = 0) = 0$
3. $\hat{F}(0 < \rho < 1) < 0$
4. $\frac{\partial \hat{F}}{\partial \rho}\big|_{\rho=1} = 0$
5. $\frac{\partial \hat{F}}{\partial \rho}\big|_{\rho=0} = 0$

Assume $E_{co} < 0$

Assume the following form of the molar free energy

$$\hat{F} = \rho^n f(\rho) E_{co} \tag{A3}$$

We can now determine a second order polynomial form of $f(\rho)$ that satisfies the criteria listed above assuming that $n >= 2$.

$$f(\rho) = a\rho^2 + b\rho + c \tag{A4}$$

$$f'(\rho) = 2a\rho + b \tag{A5}$$

$$f''(\rho) = 2a \tag{A6}$$

$$\begin{aligned}
\frac{\partial \hat{F}}{\partial \rho} &= E_{co}(n\rho^{n-1}f(\rho) + \rho^n f'(\rho)) \\
&= E_{co}\rho^{n-1}(a\rho^2(n+2) + b\rho(n+1) + cn)
\end{aligned} \tag{A7}$$

$$\frac{\partial^2 \hat{F}}{\partial \rho^2} = E_{co}\rho^{n-2}[(n+2)(n+1)a\rho^2 + (n+1)nb\rho + n(n-1)c] \tag{A8}$$

- Criteria 1: $1 = a + b + c$
  from criteria 4: $c = 1 + a + n$
- Criteria 2: guaranteed by $n >= 1$
- Criteria 3:
- Criteria 4: $n + 2a + b = 0$
  $b = -(n + 2a)$
- Criteria 5: guaranteed by $n >= 2$

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
