# Peer review of "Extending Density Phase-Field Simulations to Dynamic Regimes"

_metals, doi:10.3390/met13081497_

Round 1

Reviewer 1 Report

The proposed method is lack of validation.

Author Response

We thank the reviewer for their feedback; however, we respectively disagree with the reviewer comment that a lack of validation is not given in the manuscript. On the contrary we would direct the editor and reviewer to Figures 4-6 as sufficient evidence of simulation validity. Figure 4 illustrates that the grain boundary width vs. gradient energy coefficient curve follows a square root relationship with that relationship well established in the phase field literature and indicates that the dynamic density phase field model does not deviate from traditional phase field results. Similarly, Figure 5 illustrates the linear relationship between applied driving force and grain boundary velocity for a 1D grain boundary. Here, linearity has been a well-established outcome for grain boundaries using atomistic simulations (see the cited papers by Olmsted et al). As a result, grain boundary motion predicted by the dynamic density phase field model can be trusted in the steady state regime. Finally, the 2D shrinking grain problem is a universally accepted benchmark within the phase field community for validation. We point to the thesis by Kamachali as a good resource for more details on its derivation and use. The close agreement between the analytical relationship and the numerical model indicate that the dynamic density phase field model is accurate in transient problems as well as for multidimensional simulations.

Reviewer 2 Report

It’s very interesting and useful publication. Authors consequentially develop phase field methods. It’s seems very important that  simulation technique can be used for the different scale systems- micro and as it presented in this article - mesoscopic physics for grain boundaries. 

Nevertheless the conclusion about the influence of DPF method on performance and accuracy of simulation should have digital confirmation. It’s can be very useful for investigation of the influence the different elements\molecules on the width and morphology of grain boundaries.

Author Response

The authors appreciate the reviewer’s compliments of the manuscript and its impact. The authors would like to specific points in the manuscript that provide sufficient confirmation of the simulation performance and the accuracy that has now been improved. The increase in accuracy stems from the free energy formulation that eliminates the sharp / discontinuous first derivative of density in the grain boundary density profile (compare figure 1A and figure 3B), as well as by resolving the thermodynamic inconsistency pointed out in the theory portion of the paper. Nevertheless, the increase in performance is more subtle, but the authors do mention that to achieve similar results using a free energy expression in terms of the gradients of density, at least second order terms would be required. This leads to model A / non-conserved dynamics equation that is fourth order or higher. It can be determined directly using numerical analysis that said partial differential equations are stiffer / more computationally expensive to solve with the numerical analysis texts by Burden (Numerical Analysis) and Han (Theoretical Numerical Analysis) for further details on this point.

Reviewer 3 Report

The authors compare the DPF with the traditional phase-field method, and try to answer the three questions the authors raised. A generic DPF free energy functional is derived to carry out equilibrium and dynamic simulations of grain boundaries. The results are convincing. But the manuscript structure needs to be adjusted to highlight the research focus. The manuscript title is too broad, and the introduction is mentioned too much, which unclears the manuscript topic. Concise background introduction, more data discussion, and possible comparison with the traditional phase-field method are welcome.

 Good.

Author Response

The authors appreciate the reviewer’s critiques and summary of the manuscript and provide the following responses.

  • In order to increase the focus of the paper, the authors have changed the title to “Extending Density Phase Field Simulations to Dynamic Regimes”
  • The authors have added references to traditional phase field simulations are requested in an attempt to provide additional clarity on the changes made to the density phase field methods presented here in comparison to traditional phase field simulations.
  • We would note that other reviewers asked for additional introduction and complemented the document as a whole. The authors appreciate the reviewers request that the document be restructured but are hesitant to do so in order to avoid making changes that are not appreciated by the other reviewers.

Reviewer 4 Report

1. There are many question marks in the text, what do they mean.   Table? 22? 23

2. The literature citation is not comprehensive, and some classic and important literature is needed to introduce the origin of simulation methods

3. The correctness of the simulation cannot be determined without experimental verification

4. Is there a comparison between this simulation method and other simulation methods, and what are their advantages

5. Are there any requirements for the simulated materials that can be extended to other materials with different crystal structures

Minor editing of English language required

Author Response

The authors provide the following responses to the reviewer’s questions and comments:

  • “There are many question marks in the text, what do they mean. Table? 22? 23”
    • These are errors with the LATEX package used to generate the document. The author has removed them. We apologize for the retention of these in the original submission.
  • “The literature citation is not comprehensive, and some classic and important literature is needed to introduce the origin of simulation methods”
    • The authors have added six additional references that touch on the origin of the simulation method as well as the use of phase field simulations for simulating grain growth.
  • “The correctness of the simulation cannot be determined without experimental verification”
    • The authors point to figures 4-6 as sufficient evidence of simulation validity. This has been addressed previously with respect to Reviewer #1 and are placed below for reference. We do note that this paper is focused on computational modifications to the dynamic motion of a boundary with respect to density phase field. The addition of experimental work is beyond the scope of this particular paper, but this manuscript does set forth future work where its methodology may be applied to experiments, particularly with the validation of the modeling through standard practices and described below.

“Figures 4-6 provide sufficient evidence of simulation validity. Figure 4 illustrates that the grain boundary width vs. gradient energy coefficient curve follows a square root relationship with that relationship well established in the phase field literature and indicates that the dynamic density phase field model does not deviate from traditional phase field results. Similarly, Figure 5 illustrates the linear relationship between applied driving force and grain boundary velocity for a 1D grain boundary. Here, linearity has been a well-established outcome for grain boundaries using atomistic simulations (see the cited papers by Olmsted et al). As a result, grain boundary motion predicted by the dynamic density phase field model can be trusted in the steady state regime. Finally, the 2D shrinking grain problem is a universally accepted benchmark within the phase field community for validation. We point to the thesis by Kamachali as a good resource for more details on its derivation and use. The close agreement between the analytical relationship and the numerical model indicate that the dynamic density phase field model is accurate in transient problems as well as for multidimensional simulations.”

  • “Is there a comparison between this simulation method and other simulation methods, and what are their advantages?”
    • The Density Phase Field (DPF) method is quite young and has not garnered enough interest in its current stage of development to warrant the writing of a systematic review that compares DPF with traditional phase field simulations as well as other mesoscale simulation methods. The authors contrast the changes presented here with existing DPF methods in the theory section as well as in figures 1 and 3, as such work is needed to eventually bring DPF to full fruition and comparison to traditional phase field. Specifically, the work here aims to improve upon the current model by now enabling it to simulate moving grain boundaries, a critical next step for DPF development whereupon it can more readily be applied between research groups, increasing citations of its applicability, which would then prompt a thorough review paper of DPF with conventional phase field approaches.
    •  
  • Are there any requirements for the simulated materials that can be extended to other materials with different crystal structures.”
    • The methods illustrated in this paper are crystal structure agnostic. The thermodynamic criteria mentioned on page 5 should hold regardless of crystal type. There are free energy formulations for DPF simulations that are dependent on crystal structure (MOPF methods mentioned on page 3) but they are not the focus of this paper. Rather, we aimed to address the requirements necessary to make DPF compatible for the moving grain boundary problem, which is the next iterative step for this model’s development.

Round 2

Reviewer 1 Report

If the model can only be validated by analytical model, then why is this model useful? More validation has to be added before publication.

Author Response

We appreciate the opportunity to expand our discussion on the validation of our model. In our revised manuscript, we highlight how density phase field (DPF) has been instrumental in understanding spinodal decomposition in experimentally characterized interfaces (see references 14, 15, and 19). However, in these papers, the DPF application did not require a moving boundary. When we began to look at moving boundaries, inconsistencies in the current DPF theory / model appeared. This motivated our paper that addresses those issues while maintaining the enhanced capabilities exhibited by existing DPF models. Hence, the mode is quite useful.

This paper’s focus is on making the necessary corrections to DPF theory in order to resolve its issues with moving grain boundaries. We have submit it to the special issue in Metals on ‘multi-scale simulations of metallic materials’ as it is specifically about theory / modeling / simulation outcomes.

Our paper presents three tests that are used to validate the changes to the model. We emphasize to the reviewer that the entire results section is aimed at demonstrating the model’s fidelity and agreement with well-established physics. Figures 4 – 6 are all accepted tests in the phase field community that have been validated using atomistic calculations, experimental results, and agreement with the thermodynamic theory of interfaces. The authors rely on agreement with these tests rather than seeking to repeat the previous work of other researchers to directly prove (validate) their results using experiments. It is not clear what, if any, further validations are needed that would change this conclusion. 

We use figure 6 as illustrative example. Curvature driven motion of grain boundaries has been well established in the literature since 1948 (see C.S. Smith Introduction to Grains, phases, and Interfaces). The authors do not need to obtain metallographic samples illustrating curvature driven grain boundary motion. We rely on the plethora of data in the literature and the well-founded theory that has been built on that data which predicts that the rate of circular grain shrinkage follows equation 39 in the paper. This equation is not a model or estimate, it is rigorously derived from thermodynamics and has ample experimental justification. Similar arguments can be made for the agreements shown between our model and theory in figures 4 and 5. To emphasize this point in the manuscript, the authors have added a highlighted section of text that draws the reader’s attention to the fact that the results achieved by the modified DPF model presented in the paper match the atomistic results published by Olmsted.

Nevertheless, we do recognize the value of applying our model to various experimental situations to understand their specific conditions. But such work is outside the scope of this paper’s focus and would require well-curated data sets where the grain boundary types, their evolution with time and temperature are characterized. Such experimental studies take several months to complete. This paper and the validated model it reports provides the foundation for such future work.

With respect to the survey questions needing improvement concerning the research design appropriateness and if the methods are adequately described, there were no specific comments for us to address.  We reiterate that we have used 3 methods of validation, each well accepted in the community, and the reviewer does contend that they are incorrect. If the request is more about applying the modeling to experimental data, we appreciate that encouragement and plan to later, particularly to ensure a well characterized data set is developed. But emphasize that simply doing more validation does not change the conclusions nor is required since the 3 tests done are well accepted tests with outcomes that confirms the correctness of the modifications to the DPF model for moving boundaries.

Reviewer 3 Report

The authors only change the manuscript title. Such revision is far from enough. Concise background introduction, more data discussion, and possible comparison with the traditional phase-field method are required.

The quality of English language is okay.

Author Response

We apologize that we failed to highlight the changes to the manuscript in our revision. We have now explicitly highlighted or changed the text color for all changes or prior comments that would address concerns raised.  The only changes that go unmarked are additional references added, as no highlights can be found in the bibliography.

Here, we describe why we have structured the paper as we have done. We hope this highlights that these requests to reduce or expand certain areas in the text were done with a specific purpose and the suggestions offered by this reviewer were not disregarded.  Rather we have been meticulous in our format with specific descriptors for ease of reading and understanding.

The focus of this paper is theoretical development of the Density Phase Field (DPF) method.  As a result, it does not necessarily generate a large number of numerical results that require discussion. Rather much of the paper is spent 1) explaining existing DPF theory, 2) deriving the necessary modifications to address the issues with DPF theory, and 3) demonstrating how such modifications result in a valid thermodynamic description of stationary and moving grain boundaries.

While we appreciate the reviewer’s comment to make the introduction concise, the reason that this was not done is because DPF is a relatively new idea and we wanted to ensure that the reader has the appropriate understanding of it to appreciate the intent of the manuscript’s theory and modeling modifications that would then follow.

We added the requested comparisons of DPF to traditional phase field, where we believe such a discussion is most appropriate in the introduction when DPF is introduced. Since this paper is concerning a fix to DPF, extensive explanation and discussion of traditional phase field distracts from the focal message and outcomes of the paper, so we aimed to be concise. Furthermore, as boundary motion in traditional phase field is established, but not in DPF, the remainder of the text is devoted to develop the DPF theory corrections to enable this effect and validate that the modifications are correct.  As this slightly increased the length of the introduction, this comes back to our earlier response that we are hesitant to make sweeping changes as other reviewers positively commented on the document as a whole when balancing competing comments.

After the introduction, we summarize the current theory and its issues; we highlight what needs to be addressed; and we establish the focus of this work associated with resolving aforementioned issues.

The reviewer has asked for more data discussion. The results and discussion section of this paper is meant to demonstrate the validity of the newly developed theory. For example, we emphasize to the reviewer that the entire results section is aimed at demonstrating the model’s fidelity and agreement with well-established physics.  As discussed, these tests used are well established. As a result, they do not require much elaboration. On the other hand, the theoretical developments presented are novel, and the resulting section with respect to that development is lengthier. The authors have discussed among themselves reformatting the paper in a more traditional manner where most of the work presented is in the results and discussion section, but we feel that this would detract from the papers readability and cloud the new findings being presented, particularly in the theory development. The discrepancy between the authors formatting of the paper, and the desired format expressed by the reviewer, is simply a disagreement on the style rather than the technical validity and novelty of the work.

We hope that our highlighting the text alternations demonstrates that we have indeed made changes. And our explanation to the paper structure (particularly being a theory-modeling focus) provides context and understanding to the reviewer for the reasons why we have presented and discussed the data as we have.

Reviewer 4 Report

It can be accepted in present form.

Author Response

We appreciate the positive reception of our work and look forward to sharing this with the community as a publication in Metals. 

Round 3

Reviewer 1 Report

The reviewer appreciated the much work that the authors have provided. But I do think the validation data is too simple, which regards no experimental data. In order to convince a new model, in-house experiment or literature experimental data must be used to discuss the incremental.

Author Response

We appreciate the time of the reviewer and we do not disagree that validation is a requisite for testing a theory and model whether using one's own data sets or referenced material.

We will reiterate that we have provided this information in our prior responses and in the manuscript evident by three different methods that yield physical outcomes that agree with experimentally observed grain boundary behavior. The validation tests are well established in the phase field community for benchmarking our model, with each validation method derived from rigorous experiments and/or simulations in the literature. We have cited references in the literature that provide this evidence, but the reviewer appears not to acknowledge these references in our prior response. Furthermore, the reviewer has provided no examples of what experimental evidence is required to consider our results valid. The only option left to the authors is to perform a vast number of time consuming and expensive experiments using an array of different experimental techniques to create a well curated data set in the hopes of satisfying an ambiguous requirement.  And, regardless of these outcomes, the validation methods employed already satisfy that the model yields correct, physical outcomes based on the literature. Simply doing more does not change the conclusion drawn in the paper.

In this recent review, the reviewer's new complaint is that our validation methods are simplistic. The reviewer has provided no indication that these validation methods are inappropriate or incorrect besides describing them as "too simple." The simplicity of these tests should have no bearing on their validity. Nevertheless, we would highlight that an evaluation of the references where we cite for our grain boundary width and energy calculation validation (ref. 29 and 30) would suggest the contrary as being “too simple.” 

We believe we are at an impasse with this reviewer. We have provided a critical rebuttal that articulates that we have indeed correctly validated our model and used referenced experimental and simulation literature to support our claims that align with accepted standards in the phase field modeling community. We would ask for an editorial decision for considering the publication of our manuscript has received three affirmed, positive reviews citing our work's impact and proper outcomes. Simply doing more to satisfy an ambiguous request will not change the conclusions  that the model modifications are correct and its impact to the community. Thank you for your further consideration based on our responses to this and the prior reviews.  

Reviewer 3 Report

Accept is okay.

Author Response

We appreciate the positive response of the reviewer to accept the manuscript. We are excited to be able to share our results to the community through the journal Metals.